# Impact Mechanism and Effect of Agricultural Land Transfer on Agricultural Carbon Emissions in China: Evidence from Mediating Effect Test and Panel Threshold Regression Model

Ying Tang * and Menghan Chen

School of Public Management, Liaoning University, Shenyang 110036, China
* Correspondence: tangying5440@163.com; Tel.: +86-15004041523

**Abstract:** In order to identify the mechanism and effect of agricultural land transfer on agricultural carbon emissions, a study was conducted by analyzing the panel data of 30 provincial-level administrative regions from 2005 to 2019. Both the intermediary effect model and panel threshold regression model are applied to test the correlation between agricultural land transfer and agricultural carbon emissions, which provides some clarity on the mechanism of agricultural land transfer affecting agricultural carbon emissions and its future trends. The research results are as follows. Firstly, agricultural land transfer has a positive effect on agricultural carbon emissions, and agricultural factor input plays a mediating role between agricultural land transfer and agricultural carbon emissions. More specifically, the input of agricultural chemical elements has a positive impact on agricultural carbon emissions, while the input of agricultural machinery elements has a negative impact on agricultural carbon emissions. Secondly, under the threshold constraint of the urbanization level, the relationship between agricultural land transfer and agricultural carbon emissions is characterized by an inverted "U" shape, with a threshold value of 0.73. In view of these findings, more attention should be directed to addressing the negative impact of agricultural land transfer on the ecological environment. Furthermore, various targeted measures should be taken to reduce the ecological risk carried by agricultural land transfer, to increase the effort made on achieving the goals of agricultural carbon emission reduction, and to promote the green and sustainable development of the agricultural industry.

**Keywords:** land use; carbon emissions; intermediary effect model; panel threshold model

## 1. Introduction

When it comes to global climate warming, a significant influencing factor for it is the increase in carbon dioxide concentration in the atmosphere due to the social and economic activities of humans [1]. It is a consensus reached among the international community that various measures must be taken possibly soon to reduce carbon emissions in response to the ongoing global climate change. As the world's largest emitter of greenhouse gases, China has committed itself at the 75th United Nations General Assembly to increasing the effort made to cut down on carbon emissions, with effective policies and measures adopted to achieve the "double carbon" goal of carbon peak by 2030 and carbon neutrality by 2060. To achieve this objective, what needs to happen first is to fully understand the overall situation of carbon emissions across China. According to the relevant data, the carbon emissions from agricultural production and land use change account for nearly one fourth of the total [2]. As a large agricultural production country, China contributes about 29% to the total agricultural carbon emissions in Asia and roughly 12% to the total carbon emissions worldwide [3]. Furthermore, it continues to increase at an annual rate of 5% on average [4]. It is estimated that China's agricultural carbon emissions will increase by 30% by 2050 if there are no effective emission reduction measures taken. Obviously, agricultural production contributes significantly to the total carbon emissions in China. Therefore, in order to achieve the "double carbon" objective, it is essential to impose stringent control on

the carbon emissions arising from agricultural productions and other relevant activities. At the same time, it is necessary to promote the eco-friendly development of agricultural productions according to the national agricultural green development scheme as part of the 14th five-year plan, which requires the reduction in agricultural carbon emissions. Under this context, there have been many studies conducted by academics on agricultural carbon emissions.

In this respect, the focus of discussion is placed on the factors that affect the scale of carbon emissions. It can be calculated by using the IPCC coefficient method [5], Kaya Porter identity (KPI) method [6], carbon footprint method [7] or others. Having an incremental effect on carbon emission changes, economic scale is the main contributor to increasing carbon emissions [8,9]. Specifically, carbon emissions can be significantly affected by the increase in manufacturing output value and international trade output value in macroeconomic indicators [10]. Furthermore, population size and energy structure are another two important factors in the increase in carbon emissions [11]. The slight changes in the soil carbon cycle may also have a significant impact on the concentration of carbon monoxide in the atmosphere. However, the current technical capacity is insufficient to quantitatively allocate carbon use [12]. The increase in carbon emissions has detrimental effects on the terrestrial climate, as manifested mainly by temperature rise [13]. The utilization intensity of fossil fuels such as coal should be restricted [14], and the carbon emissions from economic activities should be reduced progressively through the popularization of clean energy and technologies, such as solar cells, biomass, hydropower and thermoelectric conversion [15,16]. Apart from that, the scale of carbon emissions should be limited in the form of trading licenses [17]. In China, agricultural carbon emissions are usually characterized by a three-stage change of "up—down—up", and there is a difference between the west and the east [18]. The areas with high total emissions concentrate in those provinces heavily reliant on the agricultural industry [19]. The total carbon emissions are jointly affected by the development of world economy and society and policy intensity [20]. There is an inverted "U" relationship existing between agricultural carbon emissions and economic growth [21], and a "U" relationship existing between environmental regulation and carbon emission efficiency [22]. In addition, the LMDI model [23], Kaya identity [24], STIRPAT model [25], geographical weighted regression model [26] and other methods can be used to conduct quantitative analysis on the influencing factors in agricultural carbon emissions. The results show that agricultural carbon emissions can be significantly reduced by agricultural production efficiency, agricultural structure, agricultural population size, agricultural technology progress and other factors [27,28].

As a market-oriented means to improve the efficiency of rural land resource allocation, rural land transfer relates to society, economy, ecology and more. However, at present, the academic research of agricultural land transfer focuses mainly on its social and economic effects [29–31], and there is little research on the ecological effects of agricultural land transfer. At the same time, to meet the "double carbon" goal and to promote agricultural green development, more attention should be paid to exploring how agricultural land transfer affects agricultural carbon emissions. With the development of agricultural land transfer market and the increase in agricultural land transfer, agricultural land circulation has made significant impact on agricultural ecology [32]. Therefore, it is of much practical significance to analyze how to reduce the ecological risk posed by agricultural land circulation while promoting the moderate-scale practice of agricultural land circulation. Based on the panel data of 30 Chinese provinces from 2005 to 2019, an intermediary effect model and a threshold model are constructed in this study based on theoretical analysis, so as to test the impact path and mechanism of agricultural land transfer on agricultural carbon emissions. Furthermore, the hypothesis is verified, which provides a theoretical reference for effectively promoting agricultural land transfer and reducing agricultural emissions.

The contributions of this study are as follows. Firstly, an intermediary effect model is adopted to test the impact mechanism of agricultural land transfer on agricultural carbon emission in China. Secondly, an analysis is conducted as to the constraints on the

relationship between agricultural land transfer and agricultural carbon emissions. Lastly, policy implications are indicated based on the empirical results for the better coordination between agricultural land transfer and agricultural carbon emission.

## 2. Agricultural Land Transfer and Agricultural Carbon Emission

### 2.1. Agricultural Land Transfer and Agricultural Production Input

In practice, the specific input mode of production as adopted by the agricultural production subject is affected by the resource endowment of factors, market price and product demand, which leads to a technology selection bias based on labor-saving technology (such as agricultural machinery) or land-saving technology (agricultural chemicals) [33,34]. Under the traditional urban–rural dual registered residence system and the policy that prohibits the circulation of agricultural land, the abundance of rural labor and the scarcity of agricultural land have jointly contributed to the resource endowment characteristics in China. Given a huge national population, land saving technology plays a vital role in improving agricultural production efficiency to make up for the defects of agricultural land resource endowment, which makes China's input of agricultural chemicals far higher than the world average. In recent years, the central government of China has issued a series of policies to promote the orderly circulation of agricultural land, effectively keep the appropriate scale of land resources, and promote the efficiency of agricultural section and increase income of farmers. Under the guidance of the national macro policies, the transfer of agricultural land has developed rapidly. According to the statistics from the Ministry of agriculture and rural sector, there was 35.9 million $hm^2$ of agricultural land in China at the end of 2018. Agricultural land is transferred among different subjects, accounting for 48.56% of the total. With the development of agricultural land transfer and the breaking of the urban-rural separation pattern in China, the magnitude of rural labor migration and non-agriculturalization continues to improve, which has a significant impact on the factor endowment structure of agricultural production in China [35,36]. For the main body of agricultural land transfer, the increase in agricultural land stock reduces the scarcity and relative price of agricultural land resources, while the continuous outflow of rural populations leads to the relative increase in labor costs. Under this context, the main body of production will adopt labor saving technologies, that is, to increase the input of agricultural machinery and reduce the input of land saving elements. As for the subject who transfers out of agricultural land, agricultural land resources will become scarcer. Therefore, the production subject will adopt land saving technology, that is, to increase the use of agricultural chemicals for the improved output level of agricultural land.

Under the agricultural land transfer policy, the agricultural land transfer in the land market has become increasingly active, thus leading to the optimization and reorganization of agricultural land resources. Through the marginal output equilibrium effect of land market [37], agricultural land will be transferred from the farmers with low production efficiency to major grain growers, professional agricultural enterprises and other modern agricultural production organizations with high production efficiency. In this way, the efficiency of agricultural land utilization can be improved. For the entities who transfer in agricultural land, the expansion of their business may increase the demand for agricultural labor. However, due to the insufficient elasticity of labor supply due to the transfer of agricultural land, it is difficult to meet the demand for agricultural labor after production scale expansion, which will motivate the production entity to invest more in agricultural machinery and equipment for productions, thus further reducing the input of agricultural chemicals [38]. In addition, the transferred entity will concentrate to connect the scattered agricultural land, which is effective in reducing the land fragmentation caused by the decentralized management of farmers. This is conducive to reducing agricultural chemicals input. In addition, since the transferred entity is advantageous in agricultural production capacity and experience, it is easier to reduce the use of traditional agricultural chemicals by applying green and low-risk production technologies [39]. On the contrary, for the entities who transfer out agricultural land, the transfer of agricultural land has reduced

the management scale of agricultural land for each entity, which moves the labor force from agricultural production to non-agricultural activities [40]. Therefore, agricultural production has the typical characteristics of concurrent operation. For these farmers, the loss of labor makes it easier to invest more agricultural chemicals for maximum profits. In addition, the stability and duration of agricultural land property rights will have a more significant impact on the investment behavior of farmers, according to the property rights theory. Due to the unstable and short-term agricultural real estate rights, farmers tend to show shortsightedness in their investments. That is to say, farmers, as "economic people", will reject the long-term investment in agricultural land, such as building irrigation and drainage facilities, improving soil quality, etc. Instead, they choose to invest a large amount of agricultural chemicals and make other short-term investments for quick profits [41]. By improving agricultural land circulation policies, the stability of agricultural land property rights can be enhanced, which will motivate farmers to abandon short-term investment for long-term investment [42,43].

*2.2. Agricultural Production Input and Agricultural Carbon Emission*

Depending on the exact form and function of agricultural input elements, the agricultural element input in agricultural land use activities can be divided into two categories: agricultural chemical element input and agricultural machinery element input. For a long time, the use of chemical fertilizers, pesticides and other agricultural chemical elements in agricultural production activities has played a major role in improving the nutrient content in agricultural soil, reducing the yield loss of crops caused by diseases, insect pests and weeds, improving grain yield and promoting the growth of agricultural economy [44,45]. Given the expanding scale of agricultural land management and the shortage of labor force, the input of agricultural chemistry such as chemical fertilizer provides an effective solution to ensuring grain output [46]. At the same time, the continuous use of agricultural chemicals has also resulted in various issues including excessive carbon dioxide emissions [47], which is more detrimental to the ecological environment. Among them, the contribution of agricultural inputs to agricultural carbon emissions is most significant [48]. The production and utilization of chemical fertilizers are the main factors affecting agricultural carbon emissions [49,50]. Such agricultural chemicals such as chemical fertilizers, pesticides and agricultural film account for about half of the total agricultural carbon emissions [51]. As for the input of agricultural machinery, agricultural machinery technology has a substitution effect on agricultural labor force, which improves the degree of specialization for agricultural productions [52,53]. With the improvement of agricultural mechanization, large-scale agricultural machinery gradually replaces the small, energy-intensive agricultural machinery in the traditional small-scale production, which to some extent curbs agricultural carbon emissions. Meanwhile, the improved level of agricultural machinery utilization significantly promotes the optimization and upgrading of industrial structure and enhances the efficiency of agricultural production, thus reducing agricultural carbon emissions.

Based on the above analysis, the following hypothesis is proposed:

Agricultural land transfer can affect agricultural carbon emissions, with the input of agricultural production materials as an intermediate variable in the impact of agricultural land transfer on agricultural carbon emissions. Among the intermediate variables of agricultural materials input, agricultural chemical factor has a promoting effect on agricultural carbon emissions, while agricultural machinery factor input has an inhibitory effect on agricultural carbon emissions.

## 3. Materials and Methods

*3.1. Analytical Methods*

(1)    Mediating effect test. In order to verify the research hypothesis proposed in this study, that is, agricultural land transfer affects agricultural carbon emissions by affecting the

input of agricultural chemical elements, the stepwise regression equation is applied to perform a mediating effect test. The design is expressed as follows [54,55]:

$$\ln TC_{it} = \theta_1 + c \ln F_{it} + control_{it} + \varepsilon_{it} \tag{1}$$

$$\ln cp_{it} = \theta_2 + a_1 \ln F_{it} + control_{it} + \varepsilon_{it} \tag{2}$$

$$am_{it} = \theta_2 + a_2 \ln F_{it} + control_{it} + \varepsilon_{it} \tag{3}$$

$$\ln TC_{it} = \theta_3 + c\prime \ln F_{it} + b_1 \ln cp_{it} + b_2 am_{it} + control_{it} + \varepsilon_{it} \tag{4}$$

where $\ln TC_{it}$ represents the interpreted variable of agricultural carbon emissions; $\ln F_{it}$ indicates the explanatory variable of agricultural land transfer; agricultural chemical factor input ($\ln cp$) and agricultural machinery input ($am$) are intermediate variable; $control_{it}$ refers to the control variable, including agricultural financial level (fsa), agricultural land resource endowment (area), agricultural population scale (popu), agricultural output value structure (pvs), and agricultural planting structure (ps); i,t represent different provinces and time, respectively; $\varepsilon$ indicates a random error term.

At the same time, it is considered by some scholars that this method has certain flaws, who suggest using more accurate methods to conduct tests. For example, the bootstrap program developed by Preacher and Hayes [56] not only shows higher test efficiency for mediation effects, but also provides a variety of test program plug-ins for complex models. For researchers, appropriate model plug-ins can be selected to suit their needs. The reported results include the stepwise regression results and the confidence interval of unbiased correction at the 95% significance level. If the confidence interval does not contain 0, it indicates that the intermediary effect exists; otherwise, this effect is non-existent. Therefore, the method as mentioned above is adopted in this study to further verify the robustness of the results about mediating effect.

(2) Panel threshold model. There may be no linearity whether in the relationship between agricultural land transfer and agricultural carbon emissions, or in the relationship between other social and economic factors and agricultural carbon emissions. Therefore, it is necessary to introduce a nonlinear adjustment mechanism to further explore the relationship between agricultural land transfer and agricultural carbon emissions. Herein, the panel threshold regression model proposed by Hansen [57] is adopted to carry out the regression analysis of agricultural land transfer and agricultural carbon emissions, with the urbanization level (the proportion of urban population in the total population) as the threshold dependent variable. The panel threshold model is expressed as follows:

$$\ln TC_{it} = \beta_0 + \alpha \ln TC_{it} + \beta_1 \ln F_{it} \times I(urban_{it} \leq \eta) + \beta_2 \ln F_{it} \times I(urban_{it} > \eta) + control_{it} + \varepsilon_{it} \tag{5}$$

where urban represents a threshold dependent variable; $\eta$ indicates the threshold value; I denotes the indicator function. In two scenarios, one being that the urbanization level falls below the threshold value ($urban_{it} \leq \eta$) and the other being that the urbanization level exceeds the threshold value ($urban_{it} > \eta$), the impact of agricultural land transfer on agricultural carbon emissions is $\beta_1$ and $\beta_2$, respectively. The threshold model can simultaneously estimate the threshold value of the urbanization level and the slope value. The significance of the threshold effect was tested, that is, the original hypothesis $H_0$; $\beta_1 = \beta_2$. If the original hypothesis is rejected, the alternative hypothesis is accepted, that is, under different urbanization levels, the impact of agricultural land transfer on agricultural carbon emissions varies significantly.

### 3.2. Variable Definition and Data Source

(1) Explanatory variable: the explanatory variable used in this study is agricultural land transfer, which refers to the transfer of land management rights to other farmers or

organizations by the farmers with land contract management rights in rural areas. According to the existing research results, agricultural land transfer is mostly replaced by cultivated land transfer indicators [58]. Therefore, the transfer area of household contracted farmland in each province is used to represent the transfer of agricultural land in each province as the explanatory variable of this study.

(2) Explained variable: the explained variable used in this study is agricultural carbon emissions, with the narrow sense of agricultural (planting) carbon emissions as the research object. It is defined as the carbon emissions generated during the use of agricultural land, mainly including the carbon emissions generated during the use of chemical fertilizers, pesticides, agricultural films and agricultural diesel, as well as the carbon emissions generated during the irrigation and tillage of agricultural land [59]. The carbon emission accounting formula is expressed as:

$$TC = \sum_{i=1}^{n} O_i = \sum_{i=1}^{n} q_i \times \rho_i \tag{6}$$

where TC represents the total agricultural carbon emission, $O_i$ indicates the carbon emission of each carbon emission form, $q_i$ denotes the quantity of each carbon emission form, and $\rho_i$ refers to the carbon emission coefficient of each form of carbon emissions. The coefficient values of this study are detailed in the research of Ding (2019).

(3) Intermediate variable: agricultural materials input. The input of agricultural materials includes the input of agricultural chemical material and that of agricultural machinery. Among them, the input of agricultural chemical elements includes various agricultural chemicals, such as chemical fertilizers, pesticides and agricultural films, all of which are inputted by the agricultural production entities in the process of crop production. Considering the difficulty in measuring the total input of agricultural chemical material, it can be found out that chemical fertilizer is one of the most important input factors in agricultural production in China, which plays a significant role in promoting grain production [60]. In the meantime, it also contributes significantly to the total agricultural carbon emissions. Therefore, the ratio of fertilizer application to crop planting area in each province is adopted to represent the input of agricultural chemical elements. Referred to as the agricultural machinery and equipment invested by farmers and other production entities in the process of crop production, agricultural machinery input can be used to indicate the level of mechanization in the process of agricultural production. In the existing research results, the total power of agricultural machinery is mostly used to represent the input of agricultural machinery. However, this index is not applicable to accurately indicate the input level of agricultural machinery. This is due to the difficulty in collecting the data on the total power of agricultural machinery at the level of farmers and the fact that the cross regional service of agricultural machinery and the socialized service of agricultural machinery are common in China. Therefore, the total power of regional agricultural machinery is unfit to fully reflect the input of agricultural machinery. Therefore, the comprehensive agricultural machine utilization rate of crop cultivation and harvest as used by the Ministry of Agriculture is adopted in this study to measure the level of agricultural mechanization. This index is the weighted average value of machine cultivation rate, machine sowing rate and machine yield.

(4) Other variables: considering that agricultural carbon emissions may be affected by other factors, other control variables are also introduced into this study, including: ① Agricultural fiscal level: Agricultural finance refers to the government's expenditure on agricultural production activities. The higher the level of expenditure, the more conducive it will be to improving agricultural technology. Furthermore, it has a significant impact on agricultural carbon emissions. In the existing studies, the proportion of fiscal expenditure spent on supporting agriculture to the total agricultural production value is often used to indicate the agricultural financial level. Since the definition of agricultural carbon emissions in this study is specific to planting

carbon emissions, the ration of the total output value of the planting industry to fiscal expenditure on supporting agriculture is used in this study to indicate the agricultural financial level of each province. ② Agricultural land resource endowment: Due to the differences in the amount of agricultural land resources in various regions, there are variations in the status and scale of agricultural production between different regions. Consequently, there are significant differences in agricultural carbon emissions between various regions. Therefore, the per capita cultivated land area of the planting industry in each province is used in this study to indicate the endowment of agricultural land resources in each province. ③ Agricultural population scale: The scale of agricultural population tends to have immediate effects on the regional structure and scale of agricultural production, thus affecting the amount of regional agricultural carbon emissions. Therefore, the number of employees in the planting industry in each province is used in this study to indicate the size of agricultural population. ④ Structure of agricultural output value: It is expressed as the ratio of the output value of planting industry to the total output value of agriculture, forestry, animal husbandry and fishery. ⑤ Agricultural planting structure: It is indicated by the ratio of the sown area of grain crops to the total sown area of crops.

The provincial panel data from 2005 to 2019 are selected for use in this study. Due to the serious lack of data in Tibet, it is excluded from the sample. Finally, 30 provincial administrative regions in mainland China are selected as the research objects. The sample data are sourced from the "China Statistical Yearbook", "China Rural Statistical Yearbook", "China rural operation and management statistical annual report", and "China Agricultural Machinery Industry Yearbook" of the corresponding years. In order to eliminate the impact of variable dimensions and ensure the stability of the data, logarithmic processing is carried out for agricultural land transfer, agricultural carbon emission and agricultural chemical element input. Table 1 lists the descriptive statistics of variables.

**Table 1.** The descriptive statistics of variables.

| Variable Name | Mean | Std. Dev. | Min | Max |
| --- | --- | --- | --- | --- |
| Agricultural land transfer (lnF) | 12.63 | 1.38 | 8.70 | 15.34 |
| Agricultural carbon emissions (lnTC) | 5.27 | 1.01 | 2.44 | 6.77 |
| Agricultural chemical element input (lnCP) | 5.82 | 0.36 | 4.72 | 6.68 |
| Agricultural machinery input (am) | 0.50 | 0.24 | 0.02 | 1.14 |
| Financial level of agriculture (fsa) | 0.39 | 0.56 | 5.72 | 1.74 |
| Agricultural land resource endowment (area) | 1.09 | 0.77 | 0.30 | 4.79 |
| Agricultural population size (population) | 4.92 | 3.59 | 0.15 | 16.98 |
| Agricultural output value structure (pvs) | 0.52 | 0.09 | 0.34 | 0.75 |
| Agricultural planting structure (ps) | 0.65 | 0.13 | 0.33 | 0.97 |

## 4. Results and Discussion

### 4.1. Regression Analysis

(1)　Benchmark regression. Table 2 shows the baseline regression results obtained for the impact of agricultural land transfer on agricultural carbon emissions. In the absence of control variables, the simple regression of agricultural carbon emissions is performed only on the transfer of agricultural land, with the estimation coefficient being significantly positive at the 1% statistical level. When control variables are introduced and fixed effects are considered for regression estimation, the estimated coefficient of agricultural land transfer remains significantly positive at the 1% statistical level. It is indicated that agricultural land transfer has a significant positive effect on agricultural carbon emissions, as does the endowment of agricultural land resources and the size of agricultural population. Conversely, the level of agricultural finance and agricultural planting structure has a significant negative effect on agricultural carbon emissions.

**Table 2.** Results of baseline regression.

|  | **lnF** | **Fsa** | **Area** | **Popu** | **Pvs** | **Ps** | **Constant** | **R$^2$** |
|---|---|---|---|---|---|---|---|---|
| Without control variables | 0.08 *** (0.01) | | | | | | 4.84 *** (0.13) | 0.23 |
| Add control variables | 0.10 *** (0.01) | −0.17 *** (0.01) | 0.16 *** (0.03) | 0.06 *** (0.01) | 0.13 (0.20) | −0.79 *** (0.13) | 4.77 *** (0.15) | 0.51 |

Note: *** is significant at the level of 1%, and Se values are in brackets.

(2)　Intermediary effect test: SPSS 25.0 software and process 4.0 macro program plug-in are applied to conduct regression analysis on the sample data. The results are detailed as follows which are showed in Table 3. In regression equation 1, the impact coefficient of agricultural land transfer on agricultural carbon emissions is 0.29, which passes the test at a significance level of 1%. That is to say, agricultural land transfer has a significant positive impact on agricultural carbon emissions. In the regression equation 2, the influence coefficient of agricultural land transfer on agricultural chemical element input is 0.03, which passes the test at the 5% significance level as well. That is to say, agricultural land transfer has a significant positive impact on agricultural chemical element input. In regression equation 3, the influence coefficient of agricultural land transfer on agricultural machinery factor input is 0.063, which also passes the test at the 1% significance level. That is to say, agricultural land transfer also has a significant positive impact on agricultural machinery factor input. In regression equation 4, the influence coefficient of agricultural land transfer, agricultural chemical element input and agricultural machinery element input on agricultural carbon emissions is 0.30, 0.79 and −0.49, respectively, all of which pass the test at the 1% significance level. That is to say, both agricultural land transfer and agricultural chemical element input have a significant positive impact on agricultural carbon emissions. By contrast, agricultural machinery element input has a significant negative impact on agricultural carbon emissions.

**Table 3.** Intermediary effect test results of agricultural land transfer on agricultural carbon emissions.

| Variables | Regression Equation (1) lnTC | | Regression Equation (2) lnap | | Regression Equation (3) Am | | Regression Equation (4) lnTC | |
|---|---|---|---|---|---|---|---|---|
|  | β | t | β | t | β | t | β | t |
| lnF | 0.29 (0.020) | 15.24 *** | 0.03 (0.01) | 2.17 ** | 0.06 (0.01) | 8.60 *** | 0.30 (0.02) | 17.08 *** |
| lnap | | | | | | | 0.79 (0.06) | 13.50 *** |
| am | | | | | | | −0.49 (0.11) | −4.30 *** |
| fsa | −0.70 (0.04) | −16.681 *** | −0.02 (0.03) | −0.78 | −0.05 (0.02) | −3.21 *** | −0.66 (0.04) | −18.25 *** |
| area | 0.15 (0.04) | 3.880 *** | −0.17 (0.03) | −5.91 *** | 0.11 (0.02) | 7.62 *** | 0.34 (0.04) | 8.99 *** |
| popu | 0.13 (0.01) | 15.655 *** | −0.01 (0.01) | −1.96 ** | −0.01 (0.00) | −3.01 *** | 0.14 (0.01) | 19.07 *** |
| pvs | −0.66 (0.26) | −2.550 ** | −0.84 (0.19) | −4.37 *** | 0.27 (0.10) | 2.67 *** | −0.13 (0.23) | 0.58 |
| ps | −0.53 (0.21) | −2.590 *** | −0.16 (0.15) | −1.04 | 0.28 (0.08) | 3.56 *** | −0.27 (0.18) | −1.53 |
| R | 0.90 | | 0.40 | | 0.69 | | 0.93 | |
| R$^2$ | 0.81 | | 0.16 | | 0.48 | | 0.86 | |
| F | 308.59 *** | | 13.96 *** | | 67.63 *** | | 349.05 *** | |

Note: **, and *** are significant at the level of 5%, and 1%, respectively, and Se values are in brackets.

From the above results, it can be concluded that agricultural land transfer exerts a partial intermediary effect on agricultural carbon emissions by affecting agricultural material input. Therefore, the first part of the research hypothesis proposed in this study is supported. Moreover, agricultural chemical factor input exerts a positive effect on agri-

cultural carbon emissions, while agricultural machinery factor input has a negative effect on agricultural carbon emissions. Therefore, the second part of the research hypothesis proposed in this study is also supported. In terms of control variables, the impact of agricultural land resource endowment and agricultural population size on agricultural carbon emissions passes the test at the significance level of 1%. Furthermore, the impact coefficient is positive, indicating the promoting effect of agricultural land resource endowment and agricultural population size on agricultural carbon emissions. As for the impact of agricultural financial level on agricultural carbon emissions, it also passes the test at the significance level of 1%. Furthermore, the impact coefficient is negative, which indicates that to a certain extent the target of agricultural carbon emission reduction can be achieved if the local government increases its support for agriculture and promotes the progress in agricultural production technology.

In order to further verify the robustness of the intermediary effect, bootstrap is used to repeatedly extract the sample data for 5000 times and the default 95% unbiased correction interval is used to test the intermediary effect. The results are shown in Table 4. The confidence interval is [0.25, 0.33] and [0.26, 0.33] for the total effect and direct effect, respectively. The confidence interval is [0.01, 0.05] for the intermediary path of "agricultural land transfer → agricultural chemical element input → agricultural carbon emission". The confidence interval is [−0.05, −0.01] for the intermediate path of "agricultural land transfer → input of agricultural machinery factors → agricultural carbon emissions". The confidence interval does not contain 0, which confirms the significance effect propagation paths.

**Table 4.** Bootstrap test results.

| Effect Propagation Path | Coefficient | SE | BootLLCI | BootULCI |
|---|---|---|---|---|
| Total effect | 0.29 | 0.02 | 0.25 | 0.33 |
| Direct effect | 0.30 | 0.02 | 0.26 | 0.33 |
| agricultural land transfer → agricultural chemical element input → agricultural carbon emission | 0.02 | 0.01 | 0.01 | 0.05 |
| agricultural land transfer → input of agricultural machinery factors → agricultural carbon emissions | −0.03 | 0.01 | −0.05 | −0.01 |

*4.2. Threshold Effect Test*

Despite agricultural factor input verified as an important medium in the impact of agricultural land transfer on agricultural carbon emissions, the mechanism of this impact may also be affected by other social and economic factors, which leads to a non-linear relationship between them. Therefore, it is necessary to introduce a non-linear mechanism into the model. There are plenty of research results showing an inverted "U" type relationship between urbanization level and environmental pollution [61,62], and agricultural land transfer has a significant impact on urbanization. Therefore, the urbanization level is taken as a threshold dependent variable in this section to analyze the impact of agricultural land transfer on agricultural carbon emissions under the context of different urbanization levels.

(1) Threshold estimate: In this study, Stata 17.0 is applied to repeatedly sample 500 times with the Bootstrap method to test the threshold effect of explanatory variables. The results are shown in Table 5. The urbanization level passes the single threshold test, but the double threshold fails the significance test. At the same time, Figure 1 shows the model likelihood ratio function diagram of the panel threshold model drawn under a single threshold to verify the threshold estimate. The critical value of the LR statistic is 7.35 at the significance level of 5%, and the LR value corresponding to the threshold value of 0.73 falls below 7.35, which is consistent with the reality.

**Table 5.** Threshold estimation test.

| Number of Thresholds | F | p | 10% | 5% | 1% | Threshold | 95% Confidence Interval |
|---|---|---|---|---|---|---|---|
| single | 44.85 * | 0.082 | 42.03 | 50.27 | 70.61 | 0.73 | [0.72, 0.75] |
| double | 16.23 | 0.642 | 42.28 | 67.54 | 89.65 | 0.83 | [0.63, 0.85] |

Note: * is significant at the level of 10%.

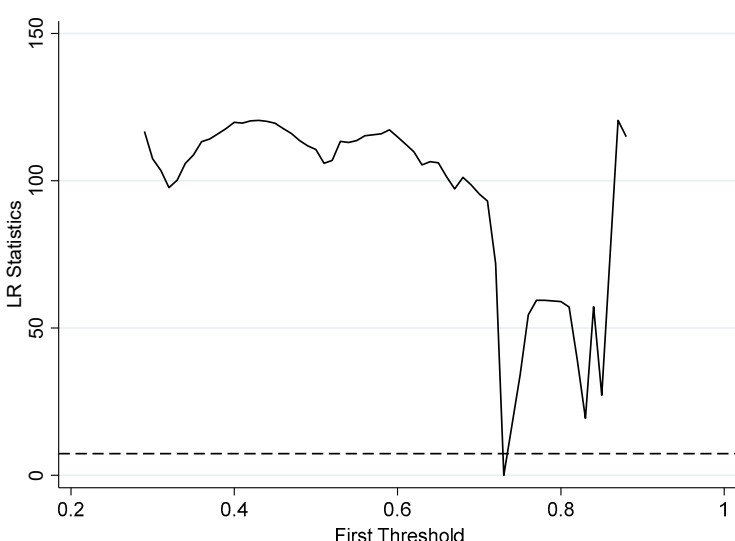

**Figure 1.** Threshold value and confidence interval of panel threshold model.

(2)  Threshold regression results. The panel threshold model is applied to analyze the sample data, with the regression results listed in Table 6. According to the results of panel threshold regression, the impact of agricultural land transfer on agricultural carbon emissions is constrained by the threshold of urbanization level. When urban $\leq 0.73$, the impact coefficient of agricultural land transfer on agricultural carbon emissions is 0.06. Agricultural land transfer exerts a positive effect on agricultural carbon emissions. Given the rapid development of rural land transfer, rural labor will concentrate in cities and towns, which improves the urbanization level. At the early stage of urbanization, rural surplus labor definitely increases agricultural capital investment to offset the loss of economic benefits caused by the outflow of agricultural labor, thus increasing agricultural carbon emissions. When urban $> 0.73$, the impact coefficient of agricultural land transfer on agricultural carbon emissions is $-0.06$. This is suspected to be due to the fact that the development of urbanization to a certain stage prompts the emergence of "anti-urbanization", as manifested in the flow of labor, capital and other factors back to the countryside, thus improving the conditions of agricultural production and driving the progress in agricultural production technology. In order to mitigate the negative external effects of agricultural production on the ecological environment, the government will also introduce the relevant environmental protection policies and regulations, which can motivate agricultural workers to improve their awareness of green production and increase the use of green and clean energy, thus comprehensively promoting the shift from traditional agricultural production to the green and efficient production characterized by "low input, high output and low pollution". Ultimately, agricultural carbon emissions are reduced. Based on the above research results, the impact of agricultural land transfer on agricultural carbon emissions shows an inverted "U" relationship under the constraint of urbanization level, which rises first and then falls. When the urbanization level exceeds a certain threshold, agricultural land transfer exerts an inhibitory effect on agricultural carbon emissions.

**Table 6.** Threshold regression results.

| Variables | lnTC | Variables | lnTC |
|---|---|---|---|
| lnF(urban ≤ η) | 0.06 *** −0.01 | pvs | −0.24 ** −0.12 |
| lnF(urban > η) | −0.06 *** −0.02 | ps | −0.06 −0.12 |
| fsa | −0.14 *** −0.01 | constant | 0.69 ** −0.31 |
| area | 0.11 *** −0.02 | R²-within | 0.69 |
| lnpopu | 0.05 *** −0.01 | F | 89.97 |

Note: **, and *** are significant at the level of 5%, and 1%.

## 5. Conclusions and Policy Recommendations

### 5.1. Conclusions

Based on China's provincial panel data of agricultural land transfer and agricultural carbon emissions from 2005 to 2019, the intermediary effect model is applied in this study to test the impact path and transmission mechanism of agricultural land transfer on agricultural carbon emissions. Furthermore, the panel threshold regression model is used to empirically test the threshold effect of agricultural land transfer on agricultural carbon emissions. On this basis, the following conclusions are drawn:

(1) Agricultural land transfer can affect agricultural carbon emissions through agricultural materials input. Specifically, agricultural chemical factor input has a positive impact on agricultural carbon emissions (0.79), while agricultural machinery factor input has a negative impact on agricultural carbon emissions (−0.49).

(2) The urbanization level exerts a significant single threshold effect on the impact of agricultural land transfer on agricultural carbon emissions. Under the threshold constraint of urbanization level, the relationship between agricultural land transfer and agricultural carbon emissions shows an inverted "U" shape. When the urbanization level falls below 0.73, agricultural land transfer exerts a promoting effect on agricultural carbon emissions. When the urbanization level exceeds 0.73, the transfer of agricultural land has an inhibitory effect on agricultural carbon emissions.

### 5.2. Policy Recommendations

(1) It is recommended to change the input structure of agricultural elements and reduce the intensity of chemical elements utilization. According to the above research results, the input of agricultural chemical elements can have a promoting effect on agricultural carbon emissions, while the input of agricultural machinery elements can exert an inhibiting effect on agricultural carbon emissions. Different management methods will have an impact on the carbon emissions from agricultural land [63]. Imposing a reasonable control on the input of agricultural chemical elements and improving the level of agricultural mechanization can reduce agricultural carbon emissions. From the perspective of the government, first, it is necessary to effectively regulate the use of agricultural chemicals at the institutional level for ensuring the agricultural ecological safety with institutional strength, including the formulation of relevant laws and regulations to agricultural carbon emissions, the establishment of a monitoring mechanism for the quality of agricultural land ecological environment, the collection of agricultural environmental taxes [64], and the increase in agricultural carbon pollution penalties. Second, the government is supposed to increase the purchase subsidies offered to farmers for using green agricultural chemicals and agricultural machinery as well as include green chemical subsidies and agricultural machinery subsidies in the ecological compensation system. This would encourage farmers to purchase green agricultural chemicals and advanced agricultural machinery [65,66]. Finally, efforts

should be made to improve the awareness of environmental protection among agricultural practitioners. This is essential for environment protection [67,68]. By publicizing the knowledge about ecological and environmental protection through mass media, the internet and other means, agricultural practitioners can better understand that the excessive input of agricultural chemicals is one of the contributors to agricultural carbon emissions. This is conducive to improving the ecological and environmental awareness of agricultural practitioners, which prompts them to reduce agricultural carbon emissions by adopting environmentally friendly technologies. From the perspective of farmers, improving the utilization efficiency of agricultural chemicals is a potential solution to reducing agricultural carbon emission. According to the survey conducted by the Ministry of Agriculture and Rural Affairs of China, the utilization rate of chemical fertilizer for grain crops in China was only 37.8% in 2017, while that of major European countries was about 65% in the same period, which indicates a significant gap. Therefore, it is worth considering the popularization of various efficient fertilization technologies such as soil testing, formulated fertilization, mechanical fertilization, planting and fertilizing, so as to reduce the amount of chemical fertilizer applied while improving the efficiency of chemical fertilizer utilization.

(2) It is suggested that the pace of urbanization can be accelerated to give full play to the inhibitory effect of high urbanization on agricultural carbon emissions. According to the above research, the impact of agricultural land transfer on agricultural carbon emissions is constrained by the threshold of urbanization level. Given the high urbanization level, agricultural land transfer exerts an inhibitory effect on agricultural carbon emissions. As for the potential negative effects of population mobility caused by agricultural land transfer, they include economic and cultural aspects [69]. Therefore, some measures may be suitable for promoting the high-quality improvement of urbanization level through agricultural land transfer. First, the government is supposed to play its role in organization and coordination, with various channels involved in the prompt delivery of employment information to farmers. Meanwhile, it is crucial to increase vocational training for farmers and improve their labor skills and overall quality. This is significant to ensuring that farmers have the ability to perform non-agricultural work and that non-agricultural labor meets market demand. Second, it is necessary to deepen the reform of the registered residence system, accelerate the unified registration and management of urban and rural household registration, promote the synchronous transformation of occupation and identity for non-agricultural employment farmers, reinforce the long-term guarantee mechanism for the citizenization of migrant workers, fully recognize the citizenship of non-agricultural employment farmers, and genuinely integrate non-agricultural employment farmers into the city. Third, the government should put in place the corresponding social security system to reduce potential risks for non-agricultural farmers [70], so as to resolve the problems encountered by the urban farmers in medical care, housing and education received by their children. In the meantime, as a basis for the survival of farmers, agricultural land resources are exposed to certain survival risks for the main body of agricultural land transfer. Therefore, it is essential to improve the effectiveness of rural social security progressively to replace the social security function of rural land, establish the employment security system for those farmers losing their land, help them to find new jobs, and solve their concerns.

**Author Contributions:** Conceptualization, Y.T.; methodology and software, M.C. All authors have read and agreed to the published version of the manuscript.

**Funding:** This study was funded by the Social Science Foundation of China (21BGL288).

**Institutional Review Board Statement:** Not applicable.

**Informed Consent Statement:** Not applicable.

**Data Availability Statement:** The sample data are sourced from the corresponding years of "China Statistical Yearbook", "China Rural Statistical Yearbook", "China rural operation and management statistical annual report", and "China Agricultural Machinery Industry Yearbook".

**Conflicts of Interest:** The authors declare no conflict of interest.

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
