# Peer review of "Impact Mechanism and Effect of Agricultural Land Transfer on Agricultural Carbon Emissions in China: Evidence from Mediating Effect Test and Panel Threshold Regression Model"

_sustainability, doi:10.3390/su142013014_

Round 1

Reviewer 1 Report

This paper explores the impact of agricultural land transfer on agricultural carbon emissions in China using mediating effects and threshold effects models. I think there is still some room for improvement in this paper.

1. The introductory section needs greater improvement. Although research on the impact of land transfer on ecological effects is relatively limited, a large body of literature has focused on agricultural carbon emissions or agricultural greenhouse gas emissions. The manuscript of this paper covers only a very small portion of this, and there is still much room for improvement in the overview of the current state of research in the literature. The following two studies on agricultural carbon emissions should be reviewed, i.e., What drives the decoupling between economic growth and energy-related CO2 emissions in China’s agricultural sector?; What drives the fluctuations of “green” productivity in China’s agricultural sector? A weighted Russell directional distance approach.

2. The hypothesis formulation appears in an odd position. The formulation of the hypothesis should correspond to the analysis of the mechanism, thus making it more relevant. Which part of the text shows that agricultural land transfer can influence agricultural carbon emissions? Please place this hypothesis explicitly after the text in that section. In addition, Hypothesis 2 is an expansion of the second half of Hypothesis 1. Please combine these two parts and place them explicitly after the analysis of the relevant mechanism.

3. The objective of the study is to investigate the impact on carbon emissions from agriculture, but the third paragraph of the introduction, which takes up a lot of space, does not mention anything about carbon emissions. I cannot understand what is the significance of this paragraph, it is more like a discussion of the context of land transfer, why is it placed after the mechanism analysis (second paragraph of the introduction)? Please make relevant changes to make it more relevant to the research topic of this paper and to improve the logic of the paragraph arrangement.

4. Please clearly state the contribution of this paper in the introduction section.

5. Please improve the quality of the language. For example, lines 95 to 101 show an excessively long sentence, please avoid this situation.

6. In section 2.1, the text in line 154 refers to the control variable "control(it)", but in formula (1), it shows "comtrol(it)". Similarly, this problem also exists in equation (2), so please double-check.

7. In section 2.2, where the selection of variables appears to refer to relevant literature, please give the specific references. For example, in the description of explanatory variables, it is mentioned that "In the existing research results, agricultural land transfer is mostly replaced by cultivated land transfer indicators."

8. Policy recommendations should be presented in a way that closely follows the conclusions of the article and is both relevant and realistic. In the Chinese context, it is debatable whether it is realistic to ask farmers to follow the concept of green development and reduce the use of agrochemicals. Putting aside the cultural context and the cost of green farming technologies, this point is clearly somewhat paper thin.

Author Response

This paper explores the impact of agricultural land transfer on agricultural carbon emissions in China using mediating effects and threshold effects models. I think there is still some room for improvement in this paper.

  1. The introductory section needs greater improvement. Although research on the impact of land transfer on ecological effects is relatively limited, a large body of literature has focused on agricultural carbon emissions or agricultural greenhouse gas emissions. The manuscript of this paper covers only a very small portion of this, and there is still much room for improvement in the overview of the current state of research in the literature. The following two studies on agricultural carbon emissions should be reviewed, i.e., What drives the decoupling between economic growth and energy-related CO2 emissions in China’s agricultural sector?; What drives the fluctuations of “green” productivity in China’s agricultural sector? A weighted Russell directional distance approach.

Response 1: We rewrite the introduction, divided it into two parts. We deleted the literature review about agricultural land transfer and added new literature review about carbon emission. New literature review includes the literatures about the relationship between the economic growth and carbon emission and the inhibitory impact factors for carbon emission.

  1. The hypothesis formulation appears in an odd position. The formulation of the hypothesis should correspond to the analysis of the mechanism, thus making it more relevant. Which part of the text shows that agricultural land transfer can influence agricultural carbon emissions? Please place this hypothesis explicitly after the text in that section. In addition, Hypothesis 2 is an expansion of the second half of Hypothesis 1. Please combine these two parts and place them explicitly after the analysis of the relevant mechanism.

Response 2: We put the hypothesis at the end of part two. Part two is a new part separated from part one to introduce the impact mechanism about agricultural land transfer on carbon emission. We also combined hypothesis one and two into one hypothesis.

  1. The objective of the study is to investigate the impact on carbon emissions from agriculture, but the third paragraph of the introduction, which takes up a lot of space, does not mention anything about carbon emissions. I cannot understand what is the significance of this paragraph, it is more like a discussion of the context of land transfer, why is it placed after the mechanism analysis (second paragraph of the introduction)? Please make relevant changes to make it more relevant to the research topic of this paper and to improve the logic of the paragraph arrangement.

Response3: We rewrite introduction and added part two about the impact mechanism analysis about land transfer on carbon emission to solve this problem.

  1. Please clearly state the contribution of this paper in the introduction section.

Response4: State the contribution at the end of the introduction of this paper.

  1. Please improve the quality of the language. For example, lines 95 to 101 show an excessively long sentence, please avoid this situation.

Response5: We improve the language of this paper with the help of native English editor.

  1. In section 2.1, the text in line 154 refers to the control variable "control(it)", but in formula (1), it shows "comtrol(it)". Similarly, this problem also exists in equation (2), so please double-check.

Response6: We revised the mistakes of variables in formula (1) and formula (2).

  1. In section 2.2, where the selection of variables appears to refer to relevant literature, please give the specific references. For example, in the description of explanatory variables, it is mentioned that "In the existing research results, agricultural land transfer is mostly replaced by cultivated land transfer indicators."

Response7: We added the references about the explanation of variables.

  1. Policy recommendations should be presented in a way that closely follows the conclusions of the article and is both relevant and realistic. In the Chinese context, it is debatable whether it is realistic to ask farmers to follow the concept of green development and reduce the use of agrochemicals. Putting aside the cultural context and the cost of green farming technologies, this point is clearly somewhat paper thin.

Response8: We deleted the countermeasures about asking farmers to follow the concept of green development and reducing the use of agrochemicals and added more available countermeasures.

Reviewer 2 Report

Review of manuscript number sustainability-1920999

General note:

The authors aimed to use the intermediary effect model and panel threshold regression model to test the impact relationship between agricultural land transfer and agricultural carbon emissions. The implementing methodology was followed to reach the goal. The results were based on hypothesis testing and analysis. Some explanation of the results should be verified. Allover, the manuscripts need further revision in terms of its structure and language.

Specific notes:

Lines 10-14: this is a running sentence >> please rewrite and merge.

In the abstracts: please add a few lines about the applied methodology.

Lines 22-25: this is a running sentence >> please rewrite and merge.

In the keywords: it is recommended to replace the first two with other keywords, as these two are already in the title.

The introduction is too long and contains general information and statements. Please rewrite and keep on the sentence that serve the goal. Meanwhile, ad two to three paragraphs on the methods that the researchers use in the previous studies and what are there pros and cons. Such as in the lines 171-179. These could be part of the introduction.

The authors use the term “this paper” many times: it is recommended to replace the word paper by “study”.

Lines 358-360: how did you reach this point?

The conclusion is some kind of a repetition of previous sentences. Please rewrite.

Author Response

Response to Reviewer 2 Comments

The authors aimed to use the intermediary effect model and panel threshold regression model to test the impact relationship between agricultural land transfer and agricultural carbon emissions. The implementing methodology was followed to reach the goal. The results were based on hypothesis testing and analysis. Some explanation of the results should be verified. Allover, the manuscripts need further revision in terms of its structure and language.

Specific notes:

  1. Lines 10-14: this is a running sentence >> please rewrite and merge.

Response 1: We improve the language of this paper with the help of native English editor.

  1. In the abstracts: please add a few lines about the applied methodology.

Response 2: We added more description about the method we chose in the abstract.

  1. Lines 22-25: this is a running sentence >> please rewrite and merge.

Response 3: We improve the language of this paper with the help of native English editor.

  1. In the keywords: it is recommended to replace the first two with other keywords, as these two are already in the title.

Response 4: We replace the first two keywords.

  1. The introduction is too long and contains general information and statements. Please rewrite and keep on the sentence that serve the goal. Meanwhile, ad two to three paragraphs on the methods that the researchers use in the previous studies and what are there pros and cons. Such as in the lines 171-179. These could be part of the introduction.

Response 5: We rewrite the introduction, divided it into two parts. We deleted the literature review about agricultural land transfer and added new literature review about carbon emission.  New literature review includes the literatures about the research methods used in previous studies.

  1. The authors use the term “this paper” many times: it is recommended to replace the word paper by “study”.

Response 6: We replaced “this paper” by “this study”

  1. Lines 358-360: how did you reach this point?

Response 7: We added short description of the software we used to do the test.

     8.The conclusion is some kind of a repetition of previous sentences. Please              rewrite.

Response 8: We rewrite the conclusions.

Reviewer 3 Report

Dear authors

Thank you for this interest research and valuable contribution.

I do not have any further recommendations. I accept the paper in this present form

Author Response

 We really appreciate for your support.

 Thank you so much!

Round 2

Reviewer 2 Report

The authors have addressed the comments properly.

Good luck.